# PROFIX: Improving Profile-Guided Optimization in Compilers with Graph Neural Networks

**Huiri Tan[1]**  **Juyong Jiang[1,2]**  **Jiasi Shen[1]** *

[1] The Hong Kong University of Science and Technology
[2] The Hong Kong University of Science and Technology (Guangzhou)
{htanaj, sjs}@cse.ust.hk, jjiang472@connect.ust.hk

## Abstract

Profile-guided optimization (PGO) advances the frontiers of compiler optimization by leveraging dynamic runtime information to generate highly optimized binaries. Traditional instrumentation-based profiling collects accurate profile data but often suffers from heavy runtime overhead. In contrast, sampling-based profiling is more efficient and scalable when collecting profile data while avoiding intrusive source code modifications. However, accurately collecting execution profiles via sampling remains challenging, especially when applied to fully optimized binaries. Such inaccurate profile data can restrict the benefits of PGO. This paper presents PROFIX, a machine learning-guided approach based on hybrid GNN architecture that addresses the problem of profile inference, aiming to correct inaccuracies in the profiles collected by sampling. Experiments on the SPEC 2017 benchmarks demonstrate that PROFIX achieves up to a 9.15% performance improvement compared to the state-of-the-art traditional algorithm and an average 6.26% improvement over the baseline machine learning models. These results highlight the effectiveness of PROFIX in optimizing real-world application profiles.

## 1 Introduction

Compilers play a critical role in transforming high-level source code into efficient machine-level instructions. Beyond basic translation, modern compilers employ numerous optimization techniques to improve program performance, reduce memory usage, and ensure faster execution [33, 37, 47, 48]. Compiler optimizations, such as loop unrolling, function inlining, and instruction scheduling, aim to exploit hardware features effectively and improve overall program efficiency [14, 17, 24, 46, 55]. Profile-guided optimization (PGO) has emerged as a

Table 1: High-level comparison of Ⓢ Sampling-based and Ⓘ Instrumentation-based PGO. Advantages are highlighted and annotated with checkmarks. Sampling-based PGO offers high performance with low overhead, without requiring intrusive modifications of code, making it well-suited for scalable deployment, but is limited in accuracy.

| Type | Efficiency | Non-Intrusive | Scalable | Accuracy |
|------|------------|---------------|----------|----------|
| Ⓢ | ✓ | ✓ | ✓ | ✗ |
| Ⓘ | ✗ | ✗ | ✗ | ✓ |

powerful optimization technique for improving program performance by incorporating runtime execution profiles into the optimization pipeline of compilers [8, 18]. By identifying frequently executed hot regions of code, PGO enables compilers to focus on optimizing the most frequently executed parts of a program, thereby significantly enhancing performance [17, 21, 23].

PGO is practiced in two main flavors: instrumentation-based and sampling-based, which are summarized in Table 1. Instrumentation-based PGO collects precise profile data by injecting counters and

---

* Corresponding author.

39th Conference on Neural Information Processing Systems (NeurIPS 2025).

logging code during compilation, but incurs substantial runtime overhead, rendering it unsuitable for production-scale applications [8, 21]. In contrast, sampling-based PGO [21, 40] leverages hardware features such as the Last Branch Record (LBR) [54] from performance monitoring units (PMUs) [36] to collect profile data with negligible overhead, and it does not require any intrusive modification to program code. These characteristics make sampling-based PGO particularly well-suited for deployment in latency-sensitive environments like data centers. Moreover, sampling-based profiles are extensively used in binary-level and post-link optimization frameworks [41, 42]. Despite its low overhead, the effectiveness of sampling-based PGO is fundamentally constrained by the accuracy of the collected profiles. Noise, missing samples, and imprecise attribution of sampling-based profiles can degrade optimization quality, ultimately resulting in suboptimal binary performance [20]. To this end, previous work has proposed to use traditional network flow algorithms, such as minimum cost flow formulation, to infer more accurate profiles [20, 62]. However, these techniques are unable to produce faithful profiles from sampled data because they rely on hand-tuned heuristics rather than learning program-specific patterns—they systematically under-estimate hot paths in sparse-sampled code, leaving much of the potential of sampling-based PGO untapped.

Recent advances in machine learning (ML) have shown great promise in code generation and compiler optimization [7, 27, 32, 35, 51, 60, 61]. ML models have been successfully applied to tasks such as automatic tuning of compilation parameters [2, 9, 68] and reinforcement learning-based register allocation [25, 59]. These successes highlight the potential of machine learning to automate and enhance traditional compiler heuristics, paving the way for more adaptive and intelligent optimization pipelines. Among recent ML techniques, graph neural networks (GNNs) [16] have been widely used in compiler optimization [3, 5, 12] due to their ability to model structured program representations. Programs are naturally formed by graphs, such as control-flow graphs (CFGs) and data-flow graphs (DFGs) [1], which encode rich semantic and structural information. By operating directly on these graph representations, GNNs can capture complex program behaviors and inform optimization decisions more effectively and accurately than traditional feature-based models [49].

In this work, we take the first step toward applying GNNs, and even ML-based approaches, to address the profile inference problem in the domain of compiler optimization. Our GNN-based model and framework bridge the gap between low-fidelity sampled profiles and the high-quality profile data required for effective sampling-based PGO. By learning to infer accurate execution frequencies from noisy and incomplete samples, our method improves the reliability of PGO and enhances overall compilation performance. The main contributions are summarized as follows:

- We propose a machine learning-guided approach to improve sampling-based profile-guided optimization, introducing a novel graph neural network architecture specifically designed for profile inference tasks.
- We develop PROFIX, an end-to-end framework that integrates a GNN-based profile inference model into the LLVM infrastructure [30], enabling accurate profile reconstruction without instrumentation, demonstrating its practicality and compatibility with real-world compilers.
- We conduct extensive experiments on widely used benchmark suites, showing that our approach significantly improves profile inference accuracy over existing methods and yields measurable performance gains in the optimized binaries.

The rest of the paper is organized as follows: Section 2 introduces the background of PGO and reviews the related work, Section 3 presents our design of the model and the PROFIX framework, Section 4 presents the experimental setup and results, and Section 5 concludes the paper.

## 2 Background and Related Work

**Profile-Guided Optimization.** Profile-guided optimization (PGO) improves code generation by allowing the compiler to observe the dynamic behavior of the program and, informed by this feedback, apply transformations such as aggressive inlining, basic block reordering, and loop unrolling, for which static analysis alone often makes suboptimal decisions [17, 24, 39, 55]. Instrumentation-based PGO embeds counters into the binary and therefore yields highly accurate profiles but introduces heavy runtime overhead, rendering it impractical for latency-critical services [8, 18, 43]. Sampling-based PGO, instead, leverages hardware performance-monitoring units to collect events with virtually no overhead, making it suitable for deployment at data-center scale [40]. However, sampling inherently introduces stochastic noise [63, 66] and sparsity into the collected profiles, which can

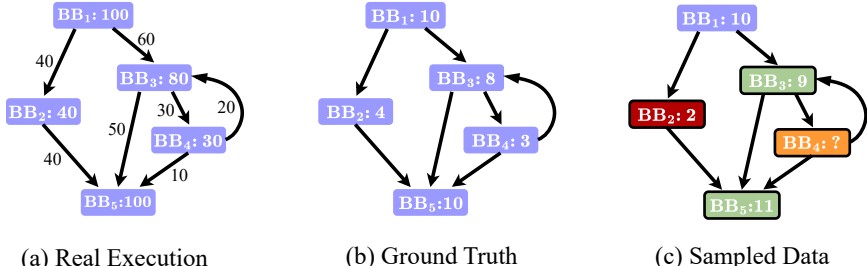

| (a) Real Execution | (b) Ground Truth | (c) Sampled Data |

Figure 1: Motivating example of the inaccuracies in sampling-based profiles. (a) The real execution count of a control flow graph, branch frequency annotated in edge. (b) Expected sampled count given the sampling rate of $\frac{1}{10}$. (c) We observe three key types of distortions. First, *underestimation* of $BB_2$, where the execution count is significantly lower than the ground truth due to insufficient sampling, which may cause the compiler to deprioritize this basic block during optimization, ultimately resulting in suboptimal performance. Second, *overestimation* in $BB_3$ and $BB_5$, where the sampled execution counts are inflated compared to the ground truth. This can mislead the compiler into making incorrect optimization decisions, such as prioritizing cold regions of the program for inlining, which may increase binary size and degrade runtime performance. Finally, the issue of a *dangling block* [20] arises with $BB_4$, which has an execution count marked as unknown, indicating that it was not sampled at all. This missing data prevents the compiler from accurately modeling control flow dependencies and can result in misaligned basic block placements or incorrect branch predictions.

mislead the compiler during optimization [20]. Consequently, the central challenge becomes profile inference, aiming at reconstructing a faithful execution profile from these incomplete, sample-driven observations so that the compiler can reap the full benefit of low-overhead PGO.

**Profile Inference Problem.** Sampling-based profiles often suffer from inaccuracies, such as underestimation, overestimation, and missing execution counts, which are illustrated in Figure 1. Such distortions stem from the probabilistic nature of hardware sampling and mislead PGO by providing unreliable execution frequency signals. Traditional solutions to this problem use the Minimum-Cost Flow (MCF) formulation [10, 31, 57], which models execution frequency inference as a flow optimization problem that minimizes the discrepancy between sampled and inferred counts while preserving flow conservation. Although effective in principle, MCF-based methods face limitations such as scalability on large and complex CFGs. To address this issue, the state-of-the-art network flow algorithm, Profi [20], introduces domain-specific heuristics and CFG transformations to accelerate MCF solving and improve practicality. However, Profi remains constrained by the rigidity of the MCF formulation, which cannot capture higher-order execution patterns or adapt to irregular control-flow structures. Moreover, its reliance on compiler-specific heuristics may lead to biased inferences when generalizing to unseen programs. These limitations motivate the need for a flexible, data-driven alternative that can generalize across program domains and scale to large workloads. In this work, we formulate profile inference as a learning problem and propose a graph-based neural approach that combines structural and sequential reasoning to infer high-fidelity execution frequencies.

**Graph Neural Networks.** Graph Neural Networks (GNNs) have emerged as a powerful tool for modeling structured data represented as graphs [34, 65, 67]. In the context of program optimization, GNNs are particularly suitable because programs can be naturally represented as various types of graphs, such as control-flow graphs and data-flow graphs [43, 44, 52]. These graph representations encode rich structural information, which GNNs can leverage to learn meaningful representations for tasks like profile inference. A classical GNN model follows the message passing paradigm [15], where node embeddings are iteratively updated by exchanging information with neighboring nodes. Specifically, at each iteration $t$, the embedding $\mathbf{h}_v^{(t)}$ of a node $v$ is updated based on the embeddings of its neighbors. The message passing process involves two key steps: *aggregation* and *update*, which can be formally described as follows:

$$\mathbf{m}_v^{(t)} = \text{AGGREGATE}^{(t)}\left(\{\mathbf{h}_u^{(t-1)} \mid u \in \mathcal{N}(v)\}\right), \tag{1}$$

$$\mathbf{h}_v^{(t)} = \text{UPDATE}^{(t)}\left(\mathbf{h}_v^{(t-1)}, \mathbf{m}_v^{(t)}\right), \tag{2}$$

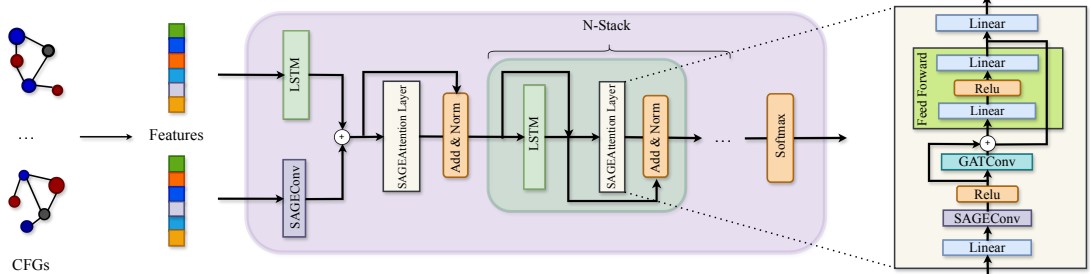

Figure 2: Overview of the model structure in PROFIX for the profile inference problem. Given a control-flow graph, we extract basic block features and process them through two parallel encoders: an LSTM for sequential dependencies and a GraphSAGE convolution for structural information. The outputs are fused and passed through stacked *SAGEAttention Layers*, each combining LSTM-based sequence modeling and attention-augmented graph reasoning. The final representation is projected through a softmax head to infer normalized execution frequencies. A zoomed-in view (right) illustrates the internal structure of the SAGEAttention Layer, which integrates GraphSAGE, GAT, and a feedforward module with residual connections and normalization.

where $\mathcal{N}(v)$ denotes the set of neighbors of $v$, and $\mathbf{h}_u^{(t-1)}$ denotes the embedding of neighbor $u$ from the previous iteration. The aggregated message $\mathbf{m}_v^{(t)}$ is combined with the current embedding of node $v$ to produce an updated embedding. Advances in GNN architectures have significantly enhanced their ability to capture complex dependencies in program graphs. Traditional message-passing neural networks (MPNNs) [15] often struggle with capturing long-range dependencies. To address these limitations, Graph Attention Networks (GAT) [58] and GraphGPS [45] introduce attention-based mechanisms that enable nodes to weigh neighboring features adaptively, allowing the model to focus on the most relevant control-flow relationships. Prior work [56] proposes a hybrid model to capture the specific pattern of directed acyclic graph, while the CFG often contains a loop. Meanwhile, GraphSAGE [19] provides an inductive learning framework that generalizes well to unseen graph structures, making it particularly suited for optimizing new, previously unseen CFGs in compiler workloads. The application of deep learning and GNNs in compiler optimization has been explored in various domains, including register allocation using reinforcement learning [25, 59], automatic loop unrolling decisions [6], and performance prediction for microarchitectural modeling [38, 53].

## 3 Methodology

The profile inference problem arises from inaccuracies in sampling-based profile-guided optimization. We next formalize the problem and present the design of the GNN model and the PROFIX framework.

### 3.1 Problem Formulation

Given a control-flow graph of a program, represented as $G = (V, E)$, where $V$ is the set of basic blocks and $E$ is the set of control-flow edges, the goal is to infer accurate execution frequencies for each basic block and edge. Each vertex $v \in V$ is associated with a sampled execution count $w(v)$, but these counts are often noisy or incomplete due to sampling errors. Formally, the task is to learn a mapping function $\mathcal{F} : G \to \mathbb{R}^{|V|}$ that predicts execution frequencies $f_i^{\text{pred}}$ for each basic block $i$, such that the inferred values closely approximate the true execution frequencies $f_i^{\text{true}}$. These inferred values reflect more realistic program execution behavior than the sampled execution counts.

### 3.2 Profile Inference Model

We present the design of our profile inference model based on graph neural networks in Figure 2. Since the natural graph structure of LLVM IR [30] includes control-flow graphs and data-flow graphs, it is particularly well-suited for modeling with GNNs. Given the numerous different CFG structures in diverse real-world programs, we aim to develop a model with strong *inductive gen-*

*eralization* capabilities to ensure the model can reliably infer execution profiles for unseen CFGs while maintaining robust performance across varying program structures. Toward this goal, we propose a hierarchical learning paradigm that alternates between *sequential learning* and *graph-based structural learning*. By alternating between these two components, our model effectively captures both execution sequences of the program and the CFG structural dependencies, enabling the strong inductive generalization capability of our model for a large variety of CFGs.

### 3.2.1  Program Feature Extraction

In PROFIX, each basic block (BB) in the CFG is represented as a feature vector $x_i \in \mathbb{R}^d$ extracted from LLVM IR. Given a CFG $G = (V, E)$, where $V$ is the set of basic blocks and $E$ represents the control-flow edges, we apply a learnable linear transformation $W_{\text{proj}} \in \mathbb{R}^{d_h \times d}$ to project each basic block feature into a latent space, following the reversed post-order traversal in CFG:

$$h_i^{(0)} = W_{\text{proj}} x_i. \tag{3}$$

These feature vectors encode both static properties (e.g., instruction count, control dependencies) and sampled profiling statistics. Details of features are available in Appendix E.

### 3.2.2  Hierarchical Representation Learning

To capture both sequential and structural dependencies within the CFG, we employ an alternating stack of LSTM layers, which model sequential dependencies, and Graph Attention layers, which model structural dependencies. This design enables our model to aggregate local neighborhood information while preserving long-range dependencies inherent in control flow.

**Sequential Encoding via LSTM.** The sequential structure of basic blocks, as reflected in their execution order, is captured by a multi-layer Long Short-Term Memory (LSTM) network [4]. The LSTM processes the sequence of basic blocks, which are extracted through a reversed post-order traversal of the control-flow graph:

$$h_i^{\text{seq}} = \text{LSTM}(h_i^{(0)}). \tag{4}$$

**Structural Encoding via GraphSAGE.** While sequential features help the model capture a single execution path of programs, real-world applications often involve complex control-flow graphs rather than flattened sequences. To encode the structural information within the CFG, we employ an inductive graph neural network that can generalize to unseen graphs. Specifically, we use GraphSAGE [19], which aggregates information from neighboring nodes to capture the structural dependencies in the CFG. The aggregation function $\text{AGG}(\cdot)$ we use in PROFIX is mean pooling, and the learned weights $W_{\text{sage}}$ are used to transform the aggregated neighborhood features:

$$h_i^{\text{sage}} = \sigma \left( W_{\text{sage}} \cdot \text{AGG} \left( \{h_j^{(t-1)} : j \in \mathcal{N}(i)\} \right) \right). \tag{5}$$

While LSTM-based encoding and GraphSAGE encode sequential and structural relationships among basic blocks, neither alone fully characterizes program behavior, which involves both *localized execution dependencies* (sequential patterns) and *global structural properties* (CFG connectivity). To address this challenge, we fuse the sequential and structural representations by concatenating their embeddings, retaining both execution order and global control-flow information.

**Enhanced Inductive Learning with Graph Attention.** As illustrated in Figure 2, to further refine the representation and enhance feature interactions, we pass fused features through a *SAGEAttention Layer*, which extends the vanilla GraphSAGE aggregation by incorporating attention mechanisms:

$$h_i^{(t)} = \text{SAGEAttention}(h_i^{\text{concat}}). \tag{6}$$

This layer combines GraphSAGE aggregation (Equation 5) with Multi-Head Graph Attention (GAT) [58], which aggregates information from neighboring nodes with attention weights:

$$h_i^{\text{gat}} = \sum_{j \in \mathcal{N}(i)} \alpha_{ij} W_{\text{gat}} h_j, \tag{7}$$

$$\alpha_{ij} = \frac{\exp\left(\text{LeakyReLU}(a^T[Wh_i \oplus Wh_j])\right)}{\sum_{k \in \mathcal{N}(i)} \exp\left(\text{LeakyReLU}(a^T[Wh_i \oplus Wh_k])\right)}. \tag{8}$$

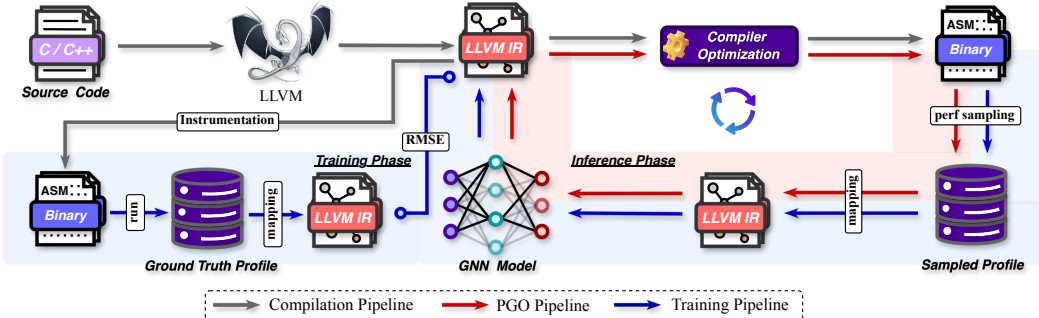

Figure 3: Overview of the PROFIX framework. The system operates in two phases: a *training phase* (blue background) and an *inference phase* (red background). In the training phase, both sampled and ground-truth profiles are collected for C/C++ programs. The sampled profile is obtained using `perf`, a hardware-based sampling tool, while the ground truth is generated via instrumentation. Both profiles are aligned with the LLVM IR [30] and mapped to control-flow graphs, which serve as the input to the model training pipeline. The trained model learns to reconstruct high-fidelity execution profiles from noisy samples. In the inference phase, only the sampled profile is required. Given a new program, we compile it to LLVM IR and extract CFGs. These CFGs are fed into the trained model to infer execution frequencies for each basic block. The inferred profiles are then passed to the PGO pipeline to guide subsequent compiler optimizations. PROFIX integrates seamlessly with the LLVM infrastructure and supports both static and post-link binary optimization workflows.

The final representation is then passed through a feedforward transformation, enabling the model to selectively focus on important nodes while preserving inductive generalization. To improve stability, the output is normalized using residual connections.

To enhance inductive generalization, we employ a stacked architecture in which LSTM layers and SAGE-Attention layers alternate. This design allows the model to iteratively refine basic block representations by incorporating both local neighborhood information and execution sequence dependencies. This stacked representation learning is repeated across $N$ layers, to enable a progressive refinement of embeddings. Finally, to infer the execution frequency distribution, we apply a softmax projection over the final representation $h_i^{(N)}$, which ensures that the predicted frequencies form a valid probability distribution, consistent with real-world execution patterns.

### 3.3 Profile Inference Pipeline

As illustrated in Figure 3, PROFIX is designed to enable accurate and scalable profile inference within modern compiler infrastructures. Unlike traditional heuristic-based solutions [20], it learns to reconstruct execution profiles directly from sampled data, effectively handling noise, missing coverage, and structural variability in real-world programs. By combining control-flow-aware graph representations with neural inference, PROFIX generalizes across diverse applications and adapts to irregular execution patterns that are difficult to model symbolically. It integrates seamlessly into the LLVM toolchain, requires no changes to compiler internals, and supports both training-time supervision and deployment-time inference. Therefore, PROFIX is highly practical for production-scale adoption, enabling high-fidelity profile-guided optimizations without the cost of instrumentation.

## 4 Experiments

We evaluate PROFIX with a diverse dataset covering compiler toolchains (Clang [29], GCC [50]), database systems (MySQL [11], SQLite [13]), and performance benchmarks (SPEC CPU 2017[2]). These applications span a broad range of code complexity, function structure, and execution behavior.

---

[2]https://www.spec.org/cpu2017/

Table 2: Dataset statistics across five representative applications used for training and evaluation. For each application, we report the total number of source functions, sampled functions (with available profiles), and filtered functions (after quality control preprocess), along with the median (q-50%) and high-percentile (q-90%) number of basic blocks per function to reflect CFG complexity. Total lines of code (LOC) are also listed to reflect program scale. The dataset includes diverse program sizes and structural complexity, providing a robust basis for learning generalizable execution profiles.

| Application | Source Funcs | Sampled Funcs | Filtered Funcs | Blocks / Function | | Total LOC |
| | | | | q-50% | q-90% | |
|---|---|---|---|---|---|---|
| Clang 20.0 | 354,365 | 4920 | 2557 | 98 | 264 | 9,192,184 |
| GCC 14.2 | 196,337 | 5120 | 1643 | 92 | 251 | 6,672,238 |
| MySQL 8.32 | 64,813 | 2467 | 1199 | 78 | 269 | 4,551,087 |
| SQLite 3 | 12,794 | 235 | 87 | 29 | 81 | 630,516 |
| SPEC CPU 2017 | 274,677 | 8696 | 575 | 153 | 611 | 13,815,845 |

## 4.1 Dataset Curation

Table 2 summarizes key dataset statistics. SPEC CPU 2017 stands out with the highest CFG complexity, underscoring its value for evaluating profile inference in highly diverse scenarios. For each benchmark, we run a representative workload twice to collect profiles, using sampling-based and instrumentation-based methods, respectively. The sampling-based profile, obtained via Linux `perf`, provides noisy execution counts and serves as the model input, while the instrumentation-based profile offers ground-truth data for supervision. All profiles are mapped back to their corresponding LLVM IR [30] CFGs to construct a profile-grounded dataset. To ensure data quality, we apply several preprocessing steps. We first filter out functions with insufficient sampling data or inconsistent basic block structures, as they lack meaningful learning signals. Consequently, we normalize execution counts to reduce sampling bias and stabilize training. Finally, we extract CFGs for each function, encoding both structure and execution profiles as node and edge attributes.

## 4.2 Implementation Details

We conduct all training and testing experiments on a server with 2×Intel(R) Xeon(R) Gold 6444Y CPU (16 Cores), 256 GB RAM, and 2×RTX 5880 GPU (48 GB Memory). Table 3 describes the partition of the collected datasets. The dataset used in the training and testing phase consists of CFGs with profiles collected from Clang, GCC, MySQL, and SQLite. The processed data is split into training, validation, and

Table 3: Overview of dataset partitioning in two experimental phases: ❶ in-distribution training and testing, ❷ out-of-distribution evaluation.

| Phase | Source | Split | Results |
|---|---|---|---|
| ❶ | Clang 20.0, GCC 14.2 MySQL 8.32, SQLite 3 | 80%, 10%, 10% (train, val, test) | Fig. 4, 5 |
| ❷ | SPEC CPU 2017 (Unseen data) | No split | Tab. 4, 5 |

test sets with a ratio of 80%/10%/10%. The training objective is formulated as a regression problem, where the predicted execution frequencies are optimized to align with the ground truth collected from instrumentation. We employ the Root Mean Squared Error (RMSE) Loss during the training phase, which measures the discrepancy between the inferred execution frequencies and the ground-truth profiles. RMSE penalizes large deviations more heavily than small ones, making it well-suited for profile inference tasks where execution frequency errors can significantly impact downstream compiler optimizations. Given a set of basic blocks $V$ in a control-flow graph, let $f_i^{\text{pred}}$ and $f_i^{\text{true}}$ denote the predicted and ground-truth execution frequencies for basic block $i$, respectively. The RMSE loss is defined as:

$$\mathcal{L}_{\text{RMSE}} = \sqrt{\frac{1}{|V|} \sum_{i \in V} (f_i^{\text{pred}} - f_i^{\text{true}})^2}. \tag{9}$$

To evaluate the accuracy of our inferred execution profiles, we employ a widely used metric, node frequency overlap [20], as the primary evaluation metric to report our performance evaluation of PROFIX on SPEC CPU 2017. Given a set of basic blocks $V$ in a control-flow graph, let $f_i^{\text{true}}$ and $f_i^{\text{pred}}$ denote the ground truth and inferred execution frequencies for basic block $i$, respectively. The

Table 4: Performance comparison of PROFIX and its ablated variants across SPEC 2017 benchmarks. Each row corresponds to a specific ablation, evaluating the impact of removing or replacing individual components, including Ⓐ the sequence encoder, Ⓑ graph reasoning modules, Ⓒ parts of the SAGEAttention layer, and Ⓓ removed input features. Results are reported in terms of overlap (%), with the highest value for each column highlighted in bold.

| Group | Variant | Benchmarks | | | | | | | | | | | | Average |
|---|---|---|---|---|---|---|---|---|---|---|---|---|---|---|
| | | perlbench | gcc | mcf | omnetpp | x264 | deepsjeng | leela | xz | namd | parest | povray | imagick | |
| Ⓐ | w/o LSTM | 89.79 | 91.62 | 96.25 | 92.93 | 92.05 | 87.20 | 93.75 | **87.57** | 89.34 | 95.04 | 92.57 | 97.03 | 92.70 |
| | w/ GRU | 92.55 | 91.90 | 91.33 | 91.99 | 93.33 | 86.61 | 94.70 | 86.16 | 88.01 | 93.56 | 90.87 | 98.68 | 92.37 |
| Ⓑ | w/o GNN | 88.37 | 90.41 | 91.75 | 89.51 | 91.20 | 88.80 | 90.98 | 80.97 | 78.48 | 91.70 | 89.74 | 95.56 | 90.48 |
| | w/ SAGE | 93.99 | 92.44 | 93.48 | 90.98 | 94.05 | **91.93** | 93.62 | 82.10 | 84.28 | 93.17 | 91.29 | 97.46 | 92.42 |
| | w/ GAT | 95.79 | 92.20 | 90.48 | 90.90 | 93.82 | 89.53 | 94.94 | 85.47 | 92.07 | 94.33 | 94.78 | 97.86 | 92.82 |
| | w/ GCN | 93.89 | 91.26 | 91.61 | 90.41 | 92.11 | 83.32 | 94.89 | 84.95 | 86.31 | 92.90 | 90.61 | 94.69 | 91.72 |
| Ⓒ | w/o SAGE | 93.19 | 91.83 | 92.96 | 91.27 | 93.39 | 85.38 | 92.98 | 83.39 | 79.54 | 92.25 | 93.12 | 91.87 | 91.36 |
| | w/o Attention | 91.14 | 90.64 | 91.51 | 91.18 | 91.78 | 90.57 | 94.41 | 78.70 | 78.27 | 93.56 | 88.53 | 90.17 | 91.29 |
| | w/o Feed-Forward | 94.92 | 92.15 | 93.72 | 91.78 | 94.04 | 88.90 | 94.69 | 78.42 | 83.05 | 93.87 | 93.00 | 95.56 | 92.75 |
| Ⓓ | Succ. Features | 88.54 | 89.42 | 89.98 | 88.57 | 89.81 | 87.19 | 90.33 | 84.91 | 83.73 | 90.57 | 89.86 | 92.24 | 90.06 |
| | Ctrl-Flow Info | 90.88 | 91.47 | 91.96 | 91.03 | 92.27 | 89.59 | 92.03 | 86.48 | 85.67 | 91.91 | 91.06 | 93.52 | 91.32 |
| | Memory Ops | 92.02 | 92.23 | 92.63 | 91.81 | 93.03 | 90.69 | 93.17 | 87.86 | 87.33 | 92.63 | 92.01 | 94.33 | 92.21 |
| | Arit./Logic Ops | 90.72 | 91.26 | 91.74 | 90.88 | 91.81 | 89.17 | 91.47 | 85.95 | 85.16 | 91.38 | 90.57 | 93.03 | 91.14 |
| | Call/Intris | 92.61 | 92.73 | 93.07 | 92.13 | 93.43 | 91.08 | 93.68 | 88.22 | 87.88 | 92.83 | 92.31 | 94.51 | 92.65 |
| PROFIX | Full Model | **95.83** | 92.35 | **96.26** | **93.98** | **94.39** | 90.78 | **95.92** | 85.74 | 88.88 | **95.12** | 92.99 | **98.85** | **93.42** |

overlap score is defined as follows:

$$\text{Overlap} = \frac{\sum_{i \in V} \min(f_i^{\text{true}}, f_i^{\text{pred}})}{\sum_{i \in V} f_i^{\text{true}}}. \tag{10}$$

This metric measures how much of the true execution profile is accurately captured by the inferred profile, with higher values indicating better inference quality. Appendix A provides a more detailed explanation about the rationality of the overlap metric.

## 4.3 Profile Inference Capability

We evaluate the profile inference capability of our model against several representative baselines under the hyperparameters listed in Appendix D, including both sequence-based and graph-based approaches, on a held-out test set. Figure 4 reports two key metrics: root mean square error (RMSE), which measures the numerical accuracy of execution frequency prediction, and overlap, which captures the alignment between predicted and ground-truth basic block execution frequencies. PROFIX achieves the best overall performance and accuracy, with an RMSE of 0.0341 and an overlap of 93.50%. These results confirm that combining sequential and structural reasoning in a unified architecture significantly improves profile inference quality.

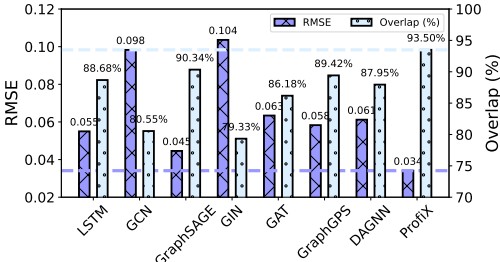

Figure 4: Performance comparison of our model and baseline methods in terms of prediction accuracy (RMSE) and consistency (overlap). Lower RMSE and higher overlap indicate better profile inference quality. PROFIX achieves the best overall result compared to all of the baselines.

## 4.4 Training Convergence Analysis

To further analyze the efficiency and stability of our approach, we compare the training and validation loss curves of all models over 60 epochs. Figure 5 presents the training loss trajectories and the validation loss curves. From the training loss curves, we observe that PROFIX converges significantly faster than baselines, achieving a lower final training loss. While GraphSAGE and GAT exhibit relatively stable convergence, their final loss remains higher, which indicates that they struggle to generalize across different CFG structures. The LSTM-based model shows slower convergence and higher training loss, which supports the hypothesis that sequential models are less effective at capturing graph-based execution patterns. In terms of validation loss, PROFIX maintains lower validation error throughout training, demonstrating strong inductive and generalization capability. In contrast, models such as GCN and GIN exhibit higher validation loss fluctuations, indicating

Table 5: The overlap (%) results on the SPEC CPU 2017 benchmarks. Symbolic represents the state-of-the-art traditional profile inference algorithm, while the rest are neural network baselines.

| Benchmarks | Symbolic | Neural Models | | | | | | | |
|---|---|---|---|---|---|---|---|---|---|
| | Profi [20] | LSTM [22] | GCN [26] | SAGE [19] | GIN [64] | GAT [58] | Graph GPS [45] | DAGNN [56] | PROFIX |
| perlbench | 86.68 | 89.76 | 77.61 | 92.31 | 77.74 | 78.74 | 91.85 | 90.37 | **95.83** |
| gcc | 91.90 | 89.23 | 82.44 | 91.95 | 82.80 | **93.39** | 90.27 | 88.93 | 92.35 |
| mcf | 95.97 | 92.31 | 89.94 | 84.93 | 93.50 | 91.48 | 94.12 | 92.06 | **96.26** |
| omnetpp | 93.76 | 87.72 | 82.52 | 92.37 | 82.34 | 80.04 | 89.18 | 88.44 | **93.98** |
| x264 | 93.12 | 94.01 | 85.47 | 94.13 | 86.20 | 86.82 | 92.49 | 91.67 | **94.39** |
| deepsjeng | 87.89 | 91.70 | **94.91** | 91.43 | 94.15 | 92.97 | 93.40 | 92.55 | 90.78 |
| leela | 92.89 | 90.77 | 86.03 | 94.92 | 86.02 | 83.96 | 92.08 | 91.30 | **95.92** |
| xz | **91.02** | 80.03 | 82.51 | 85.91 | 81.08 | 82.35 | 85.64 | 84.73 | 85.74 |
| namd | **93.84** | 87.38 | 65.38 | 77.81 | 64.15 | 69.63 | 83.25 | 81.42 | 88.88 |
| parest | 91.39 | 89.95 | 86.32 | 94.05 | 86.33 | 83.54 | 91.77 | 90.90 | **95.12** |
| povray | 92.48 | 87.43 | 82.54 | 92.00 | 81.78 | 83.43 | 90.09 | 89.67 | **92.99** |
| imagick | 90.14 | 87.69 | 75.77 | 93.83 | 75.58 | 81.95 | 94.10 | 93.15 | **98.85** |
| Average | 91.76 | 89.00 | 86.62 | 90.47 | 82.64 | 84.03 | 89.42 | 87.95 | **93.42** |

overfitting to specific function structures. Notably, LSTM struggles with stability, which indicates its difficulty in adapting to programs with CFG dependencies.

## 4.5 Ablation Study

To demonstrate the contribution of each architectural component, Table 4 presents an ablation study, where key modules in our model are systematically removed or replaced.

**A Variants on sequence modeling.** We evaluate the role of sequential encoding by modifying the LSTM-based encoder. Removing the LSTM entirely results in notable accuracy degradation, indicating the importance of capturing execution order among basic blocks. Replacing it with a simpler GRU unit also leads to performance drops, though less severe, suggesting that the higher expressive capacity of LSTM offers benefits in modeling sequential dependencies.

**B Variants on graph reasoning.** To examine the impact of structural modeling, we replace the GNN components with various alternatives. Removing all GNN layers and using a Multi-Layer Perceptron significantly impairs performance, confirming the importance of graph-based reasoning. Among inductive GNN variants, substituting our SAGEAttention layer with Graph-SAGE or GAT leads to significant degradation. In contrast, using a transductive GCN model, which is unable to generalize to unseen graphs, results in consistently lower accuracy.

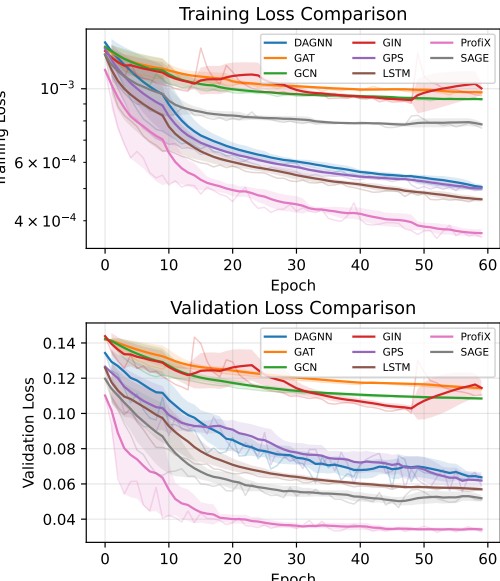

Figure 5: Training and validation loss curves for PROFIX and baseline models over first 60 epochs. Our model converges faster and achieves lower final loss on both training and validation sets, indicating superior inductive generalization capability.

**C Variants on SAGEAttention components.** We further dissect the SAGEAttention layer by ablating individual components. Removing GraphSAGE aggregation, the attention mechanism, or the feedforward transformation each causes measurable performance declines, highlighting their complementary roles in enabling expressive and context-aware message passing.

**D Variants on Input Features.** The feature ablation results show that all feature groups contribute to the performance. Removing *successor* or *control-flow* features causes the largest accuracy drop, confirming their importance for capturing execution dependencies and CFG structure. Other groups, such as *memory* and *operation* features, have smaller effects, indicating that the model can compensate for them through learned relational reasoning. Overall, the results highlight the complementary nature of our feature design and the robustness of PROFIX when features are partially removed.

Table 6: Parameter-matched comparison of baselines. All models have comparable parameter counts ($\sim 1.67$M) and are trained for the same number of epochs. Results are reported in overlap (%).

| Model | #Params (M) | Train Time (s/epoch) | Inference Time (ms/sample) | Overlap (%) |
|---|---|---|---|---|
| LSTM | 1.68 | 2.14 | 0.97 | 85.42 |
| GCN | 1.67 | 0.62 | 1.33 | 76.87 |
| GIN | 1.68 | 0.69 | 1.03 | 78.76 |
| GAT | 1.65 | 0.81 | 1.92 | 83.79 |
| GraphSAGE | 1.68 | 0.75 | 1.12 | 88.97 |
| GraphGPS | 1.67 | 1.39 | 1.97 | 89.31 |
| DAGNN | 1.66 | 1.29 | 1.79 | 86.54 |
| PROFIX | 1.67 | 1.33 | 1.72 | **93.50** |

Overall, the ablation results validate the effectiveness of our hierarchical design. The full model achieves the best performance across benchmarks, underscoring the importance of jointly modeling both CFG structure and sequential execution order for accurate profile inference.

### 4.6 Controlled Compute and Parameter Scaling for Baselines

In order to further control for the impact of model capacity and computational cost, we conducted an additional set of experiments in which all baselines were adjusted to have approximately the same number of parameters ($\sim 1.67$M), the results are summarized in Table 6. Each baseline model was first trained using the hyperparameter settings recommended by its original authors, which are widely recognized as optimal or near-optimal for the respective architectures.

Overall, our method consistently outperforms all baselines even under strictly controlled compute and parameter budgets. These results indicate that the observed performance gains stem from the architectural design and inductive reasoning capability of our model, rather than from increased model size or computational advantage.

### 4.7 Performance on Real-World Applications

To evaluate the inductive generalization capability of our model, we measure profile inference accuracy on the diverse SPEC CPU 2017 benchmarks, which feature a wide range of control-flow complexities, function sizes, and execution patterns. As shown in Table 5, our model consistently outperforms all baselines across benchmarks, achieving up to 98.85% profile accuracy on imagick, 96.26% on mcf, and 95.83% on perlbench. It shows particular strength on structured control-flow workloads like parset and leela, where traditional GNNs and LSTM struggle to deal with. Overall, PROFIX improves overlap by up to 9.15% over Profi and achieves an average 6.26% improvement over other ML baseline models, highlighting the effectiveness of our model design.

### 4.8 Impact on Downstream Compiler Optimizations

To demonstrate the practical benefits of improved profile inference, we integrate our profile inference model into the LLVM profile-guided optimization pipeline. We evaluate the performance improvements of PGO on MySQL by using profiles inferred by our model, compared to raw sampling-based and Profi-inferred profiles. Incorporating our refined profiles, the PGO-optimized binary achieves a 12.1%($\pm$0.3%) average speedup over the native binary, while raw profiles only yield an 8.3%($\pm$ 0.1%) speedup, and Profi-inferred profiles result in a 9.7%($\pm$ 0.3%) speedup. We run MySQL benchmarks multiple times to ensure these results are within 95% confidence interval under t-distribution [28]. These results highlight the effectiveness of PROFIX in enhancing PGO performance.

## 5   Conclusion

This paper presents PROFIX, a novel graph neural network (GNN)-based model and framework to improve the profile inference accuracy for sampling-based profile-guided optimization (PGO) in compilers. Comprehensive experiment results on real-world benchmark applications demonstrate that our GNN-based model achieves significant improvements over both state-of-the-art traditional algorithms and baseline machine learning models. These results demonstrate the superiority of PROFIX in improving the performance of sampling-based PGO and compiler optimization.

## Acknowledgments and Disclosure of Funding

We thank the anonymous reviewers for their insightful and helpful comments. This work was supported in part by the Hong Kong Research Grants Council (Project No. 26216025) and the Alibaba Group (through the Alibaba Innovative Research Program). The views expressed in this work are those of the authors and do not necessarily reflect the views of the funding agencies.

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

# A    Overlap Metric

To evaluate the quality of predicted execution profiles, we adopt the *overlap* metric, a widely used evaluation measure in prior work [20]. Overlap quantifies how well the predicted execution frequency distribution aligns with the ground truth. It is formally defined as:

$$\text{Overlap} = \frac{\sum_{i \in V} \min(f_i^{\text{true}}, f_i^{\text{pred}})}{\sum_{i \in V} f_i^{\text{true}}}, \tag{11}$$

where $f_i^{\text{true}}$ and $f_i^{\text{pred}}$ denote the normalized execution frequencies of basic block $i$ in the ground truth and predicted profiles, respectively.

This metric captures the intersection between predicted and actual execution distributions. An overlap score of 1 indicates perfect prediction (i.e., $f_i^{\text{pred}} = f_i^{\text{true}}$ for all $i$), whereas a score of 0 indicates no alignment at all. Note that, because the numerator uses the minimum of $f_i^{\text{true}}$ and $f_i^{\text{pred}}$, overestimating a block's frequency does not artificially inflate the score.

To ensure that $f^{\text{pred}}$ represents a valid probability distribution, our model applies a softmax layer across all basic blocks in each function:

$$\sum_{i=1}^{n} f_i^{\text{pred}} = 1,$$

where $n$ is the number of basic blocks. This normalization constrains the model to distribute a fixed amount of probability mass. As a result, increasing the predicted frequency for one block necessarily decreases the predicted frequency for others, which prevents arbitrary inflation of overlap scores.

We illustrate the overlap metric with a simple example. Suppose a function contains three basic blocks, and the true normalized execution frequencies are:

$$f^{\text{true}} = [0.5, \ 0.3, \ 0.2].$$

**Case A** (Perfect Prediction):

$$f^{\text{pred}} = [0.5, \ 0.3, \ 0.2] \quad \Rightarrow \quad \text{overlap} = 1.0$$

**Case B** (Overestimate $BB_2$, Underestimate $BB_1$):

$$f^{\text{pred}} = [0.3, \ 0.5, \ 0.2] \quad \Rightarrow \quad \text{overlap} = 0.3 + 0.3 + 0.2 = 0.8$$

As seen above, although $BB_2$ is overestimated, the increase comes at the expense of BB1, whose prediction is now lower than the ground truth. Because the overlap only accounts for the *minimum* between predicted and true values, misalignment in any direction—over or under—will reduce the score.

In summary, the overlap metric provides a meaningful and bounded measure of prediction quality. It encourages models to align with ground-truth distributions holistically, rather than simply boosting predictions for a few hot blocks. Thanks to softmax normalization, our model inherently avoids degenerate solutions where predicted frequencies are uniformly inflated, ensuring that the overlap metric faithfully reflects profile inference accuracy.

# B    Model Complexity

We analyze the theoretical computational complexity of each major component in PROFIX, as summarized in Table 7. The LSTM layer has a complexity of $O(4Nh^2 + 4Ndh)$, where $N$ is the number of basic blocks, $d$ is the input dimension, and $h$ is the hidden size. This complexity accounts for the computation of four internal gates per time step.

For structural encoding, the GraphSAGE layer incurs $O(N(kd + d^2))$ complexity, where $k$ is the average number of neighbors per node. Our SAGEAttention layer combines attention-based edge weighting and neighborhood aggregation, resulting in $O(Ed + N(kd + d^2))$, where $E$ denotes the number of edges.

The total complexity across all $L$ stacked layers is approximately $O(L \cdot (Nh^2 + Ed + Nd^2))$, reflecting the alternating design of LSTM and SAGEAttention blocks. This hybrid architecture strikes a balance between expressiveness and scalability, making it feasible for large-scale control-flow graphs extracted from real-world software.

Table 7: Complexity of different model components.

| Layer Type | Complexity | Comment |
|---|---|---|
| LSTM | $O(4Nh^2 + 4Ndh)$ | Every time step calculates 4 gates (input, forget, output, candidate), sequence length is $T = N$ |
| GraphSAGE | $O(N(kd + d^2))$ | Neighbor aggregate $N \cdot k \cdot d$ and linear transformation $N \cdot d^2$ |
| SAGEAttention | $O(Ed + N(kd + d^2))$ | GAT attention calculation $E \cdot d$ and GraphSAGE $O(N(kd + d^2))$ |
| Overall | $O(L \cdot (Nh^2 + Ed + Nd^2))$ | Every layer contains LSTM and SAGEAttention, $L$ layers, $E \sim O(kN)$ |

where

- $N$ is the number of nodes (basic blocks),
- $E$ is the number of edges,
- $d$ is the input dimension,
- $h$ is the hidden dimension of LSTM,
- $k$ is the average number of neighbors in GraphSAGE,
- $L$ is the number of our SAGEAttention layers.

## C   Training Time Comparison

We report the training, validation, and inference time for PROFIX and baselines in Table 8. Runtime is measured on a per-epoch basis for training and validation, and per-sample basis for inference. Among all methods, our model incurs the highest computational cost due to its stacked architecture of LSTM and SAGEAttention layers. Despite this overhead, the training time remains under 1.5 seconds per epoch, and inference requires only 1.72 milliseconds per sample, making it suitable for offline optimization scenarios.

Table 8: Training, validation, and inference time comparison.

| Model | Training Time (S/epoch) | Validation Time (S/epoch) | Inference Time (ms/sample) |
|---|---|---|---|
| Profi | N/A | N/A | 16.12 |
| LSTM | 1.075 | 0.258 | 0.32 |
| GCN | 0.172 | 0.276 | 0.48 |
| GraphSAGE | 0.209 | 0.235 | 0.39 |
| GIN | 0.181 | 0.180 | 0.25 |
| GAT | 0.252 | 0.388 | 0.71 |
| PROFIX | 1.328 | 0.830 | 1.72 |

In terms of computational complexity, our model combines the costs of recurrent and graph-based modules. The LSTM contributes $O(Nh^2)$ per sequence, while the SAGEAttention layer—composed of GraphSAGE and GAT—adds $O(Ed + Nd^2)$ per graph layer, where $N$ is the number of basic blocks, $E$ the number of control-flow edges, $h$ the LSTM hidden size, and $d$ the embedding dimension. Given $L$ stacked layers, the total forward-pass complexity becomes $O(L \cdot (Nh^2 + Ed + Nd^2))$. Although this hybrid design introduces greater computational cost, it yields significantly improved accuracy, as shown in earlier evaluation.

As shown in the table, the traditional algorithm does not require training but requires a longer inference time. This is due to the fact that it directly operates on the internal data structures of

LLVM, which introduces extra execution time. On the other hand, PROFIX does require more training time compared to simpler machine learning models like LSTM or GraphSAGE. However, the additional time investment leads to significant improvements in the overall performance of the optimized binaries, as discussed in Section 4.7 We attribute this difference to the tradeoff between time and accuracy, where the additional training time enables the model to generate more accurate profiles, ultimately enhancing the effectiveness of profile-guided optimization.

## D    Hyperparameters

We summarize the key hyperparameters used during model training in Table 9. The model is optimized using the Adam optimizer with an initial learning rate of 0.001, and a StepLR scheduler is applied to gradually reduce the learning rate during training. The LSTM hidden size is set to 256, and we stack three SAGEAttention layers with a dropout rate of 0.1 for regularization. Training is performed with early stopping based on validation loss, using a patience of 10 epochs. We employ root mean square error (RMSE) as the loss function, and split the dataset into 80% training, 10% validation, and 10% testing. These choices balance performance and generalization while ensuring stable convergence across benchmarks.

Table 9: Key hyperparameters for model training.

| Hyperparameter | Value |
|---|---|
| Learning Rate | 0.001 |
| Train Batch Size | 128 |
| Validate/Test Batch Size | 1 |
| Optimizer | Adam |
| Weight Decay | 0 (No weight decay) |
| Learning Rate Scheduler | StepLR (Step Size: 5, Gamma: 0.97) |
| Epochs | 300 |
| LSTM Hidden Size | 256 |
| SAGE Attention Layers | 3 |
| Dropout Rate | 0.1 |
| Early Stopping Patience | 10 |
| Loss Function | RMSE Loss |
| Train-Validate-Test Split Ratio | 80%, 10%, 10% |

## E    Key Features extracted from LLVM IR

Table 10 summarizes the key features we extracted from the LLVM IR code, based on the data collected during profiling and instrumentation. These features are essential for understanding the behavior of basic blocks in the functions being analyzed and for representing the program characteristics. Furthermore, the use of LLVM IR makes PROFIX agnostic to the source language.

In our analysis, we not only extract various instruction-level features from each basic block but also consider the structural relationship between basic blocks by incorporating the features of their successors. This design enables us to capture how the control flow and branching behavior of a program influence its execution patterns.

For each basic block (BB), the following features are computed:

- Successor Blocks: The set of successor blocks (or target blocks for branches) is considered a key feature of the current basic block. This feature is important because the control flow from one basic block to another directly impacts the execution frequency and the optimization strategies employed by the compiler.

- Successor Features: For each successor block of a given basic block, we append its own feature vector to represent the relationship between the current block and its successors. Specifically:

Table 10: Key features extracted from the LLVM IR.

| Feature Type | Fields | Description (Information Modeled) |
|---|---|---|
| **Control Flow** | `phiNumber` | Number of PHI nodes (Models control flow dependencies) |
| | `successorNumber` | Number of successors (branches) (Models control flow) |
| | `predecessorNumber` | Number of predecessors (Models control flow) |
| | `isEntry` | Flag indicating if the basic block is the entry block (Models entry point) |
| | `isExit` | Flag indicating if the basic block is an exit block (Models exit point) |
| | `isLoopHeader` | Flag indicating if the basic block is a loop header (Models loop structure) |
| **Memory Operations** | `loadNumber` | Number of load instructions (Models memory read operations) |
| | `storeNumber` | Number of store instructions (Models memory write operations) |
| | `gepNumber` | Number of GetElementPtr instructions (Models memory access) |
| | `instructionNumber` | Total number of instructions (Models overall memory instructions) |
| **Arithmetic and Logic** | `arithmeticNumber` | Number of arithmetic operations (Models computational complexity) |
| | `logicalNumber` | Number of logical operations (Models computational complexity) |
| | `castNumber` | Number of cast instructions (Models type conversion) |
| **Call and Intrinsics** | `callNumber` | Number of call instructions (Models function call behavior) |
| | `intrinsicNumber` | Number of intrinsic function calls (Models specialized operations) |

- For conditional branches, the two successor blocks corresponding to the true and false branches are treated as two distinct features.
- For unconditional branches, only the single successor block is considered.
- For switch statements, each potential successor (case or default block) is included in the feature set, with the average feature vector being calculated across all case successors.

- Feature Representation: The feature vector of each basic block includes the features of its immediate successors. This feature helps to model the impact of branching on the execution flow, where the behavior of a block's successors (such as their instruction count, branch direction, or loop presence) is critical for making predictions on the block's behavior and its contribution to the overall function execution.

- Adjacency Information: The adjacency between basic blocks is stored as a set of features that reflect the program's control flow graph (CFG). This adjacency information is concatenated with the instruction-level features of the basic block to provide a complete representation of the block in the context of its neighbors in the program's control flow. This feature includes the number of predecessors and successors, which informs the model about the block's position in the program's flow.

# F    Limitations

While our approach demonstrates strong performance across a range of benchmarks, there are a few limitations worth noting. First, the current model is trained and evaluated on statically compiled programs with well-defined control-flow graphs; applying the method to highly dynamic or just-in-time compiled workloads may require additional adaptations. Second, although our framework generalizes across multiple applications, domain-specific fine-tuning could further improve performance in specialized settings such as embedded systems or GPU kernels. Lastly, our evaluation primarily focuses on offline optimization scenarios; exploring integration with online or interactive profiling workflows remains an interesting direction for future work.

# G    Broader Impacts

Our work on enhancing compiler optimization through improved profile inference using a Graph Neural Network (GNN)-based model has several broader impacts. First, by increasing the accuracy of profile data, our approach can lead to the generation of more efficient binaries, potentially reducing energy consumption and computational costs across a wide range of applications. This is particularly significant given the growing environmental concerns associated with high-performance computing and data centers. Furthermore, the improved efficiency and performance of software may contribute to enhanced user experiences and enable more complex computational tasks to be performed on existing hardware, extending the lifespan of devices and reducing electronic waste. However, the increased

efficiency and optimization capabilities also raise concerns about potential misuse. Optimized binaries could be used to amplify the effectiveness of malicious software, making it more difficult to detect and counteract. Therefore, our work underscores the importance of incorporating safety and ethical considerations into the development and deployment of compiler optimization technologies.

