# OpenReview forum: "ProfiX: Improving Profile-Guided Optimization in Compilers with Graph Neural Networks"
_NeurIPS.cc/2025/Conference — NeurIPS 2025 poster_

### Official Review · Reviewer_Z1bD · 2025-06-25

**Clarity:** 3
**Significance:** 3
**Originality:** 2
**Rating:** 5
**Confidence:** 4

**Summary:**

This paper tackles the inaccuracy of sampling-based profiles in Profile-Guided Optimization by framing profile inference as a learning problem on program graphs. It introduces a novel GNN architecture that merges an LSTM-based sequential encoder with GraphSAGE and graph-attention layers to capture both execution order and control-flow structure. The framework is integrated end-to-end within the LLVM PGO pipeline, training the GNN using sampled and instrumentation-based profiles with LLVM IR, and deploying it without intrusive instrumentation. This approach demonstrates significant empirical gains.

**Questions:**

A few questions and areas of improvement:
Q: Can the approach handle JIT-compiled code (e.g., LLVM JIT) or dynamic languages?
Discussing how the model might be adapted or retrained for JIT scenario would strengthen the paper’s significance.


Q: Which LLVM IR features (Table 8) are most importance?
Include an ablation where you remove groups of features to show their relative importance. If certain features are redundant, removing them could reduce model size/latency without hurting accuracy.

**Ethical Concerns:**

["NO or VERY MINOR ethics concerns only"]

**Final Justification:**

the rebuttal addressed concerns about the applicability and of the work by clarifying and acknowledging the use cases or potential adaptations for JIT-compiled and dynamic languages.

**Limitations:**

yes

**Quality:**

3

**Strengths And Weaknesses:**

Strengths are:
-well-structured presentation and clear problem motivation
-addresses a practical bottleneck in industry-scale PGO and demonstrate end-to-end speedups integrated to LLVM, showing real-world impact

Weaknesses are:
-the main text could better summarize which static/dynamic features are most critical, which are in an appendix
-scope is limited to statically compiled C/C++; applicability to JIT-compiled or highly dynamic languages is to be explored

---

> ### Author Rebuttal · Authors · 2025-07-31
>
> We appreciate the feedback from the reviewer.
>
> > Include an ablation where you remove groups of features to show their relative importance.
>
> We conducted experiments on SEPC CPU 2017 where we removed groups of input features described in Appendix E to assess their impact on model performance.
>
> We categorized the input features into five groups:
>
> 1). Successor features, where each basic block’s input vector is augmented with its successors’ features,
>
> 2). Control-flow features (e.g., number of predecessors/successors, loop headers),
>
> 3). Memory operation counts (load/store/gep),
>
> 4). Arithmetic and logic operations,
>
> 5). Call and intrinsic counts.
>
> **Table 1.**
>
> | **Feature Group Removed** | **Overlap (%)** | **RMSE** | **$\Delta$ Overlap vs Full Model** |
> | ------------------------- | --------------- | -------- | ---------------------------------- |
> | **None (Full Model)**     | **93.42**       | 0.034    | —                                  |
> | 1)                        | 90.06           | 0.044    | ↓ **3.36**                         |
> | 2)                        | 91.32           | 0.039    | ↓ 2.10                             |
> | 3)                        | 92.21           | 0.037    | ↓ 1.21                             |
> | 4)                        | 91.14           | 0.041    | ↓ 2.28                             |
> | 5)                        | 92.65           | 0.036    | ↓ 0.77                             |
>
>
> Our ablation results show that removing successor features leads to the largest drop in accuracy, indicating their critical role in modeling data/control dependencies. Control-flow and memory features also contribute meaningfully, while removing compute or call features has more moderate impact.
>
>
> These results confirm that the feature design -- particularly the inclusion of successor semantics — is an important contributor to the model’s overall accuracy.
>
>
>
> > Which LLVM IR features (Table 8) are most important?
>
> The feature ablation study (see Table 1 above) shows that that successor features, arithmatic-related features and control-flow related features are the most impactful, with the removal of successor features resulting in the largest performance drop. This highlights their importance in modeling control dependencies and flow-sensitive behavior.
>
>
>
> > Can the approach handle JIT-compiled code (e.g., LLVM JIT) or dynamic languages?
>
>
> In principle, our approach is compatible with JIT-compiled code and dynamic languages, since it only requires access to the control-flow graph (CFG) and sampled or instrumented execution profiles. If the JIT compiler can emit an analyzable IR (e.g., LLVM IR) and profiling data, our method can be applied.
>
>
> That said, the deployment context differs significantly from ahead-of-time (AOT) compilation. In dynamic language runtimes (e.g., JavaScript, Python, JVM-based languages), profiling and optimization are often tightly integrated into the runtime system itself. These systems typically use internal instrumentation or sampling during interpretation or early-stage JIT to collect hot path information, which is then used to guide tiered JIT compilation.
>
> Examples:
>
> - JVM HotSpot collects method and branch execution counts during interpretation, and uses this to optimize and inline hot methods at runtime.
>
> - V8 (JavaScript engine) performs on-stack replacement and dynamic deoptimization based on inline caches and runtime type feedback.
>
> - PyPy uses tracing JIT, where hot loops are detected and compiled based on interpreter-driven profiling.
>
>
> In these systems, the runtime effectively performs its own version of profile-guided optimization (PGO) using dynamic profiling feedback. While our model could, in theory, be incorporated as a learned profile inference component within such systems, doing so would require:
>
> - Access to the JIT-compiled IR,
>
> - Alignment of runtime profile data with basic blocks or execution units,
>
> - Integration with the runtime’s internal optimization pipeline.
>
>
> We view this as a promising future direction, and our method’s inductive generalization and reliance only on control-flow graphs and noisy profiles make it a strong candidate for such integration.We will clarify this point in the final version and highlight it as a potential application scenario in dynamic environments.

---

> ### Comment · Reviewer_Z1bD · 2025-07-31
> **Thank for clarification**
>
> Thank you authors for the rebuttal. The clarifications given above and incorporation of the feedback improves the paper, thus the score should increase

---

> ### Author Response · Authors · 2025-08-01
>
> Thank you for your thoughtful feedback and kind words. We truly appreciate your time and consideration.
>
> Best regards,
>
> Authors

---

### Official Review · Reviewer_d7mA · 2025-06-29

**Clarity:** 2
**Significance:** 2
**Originality:** 2
**Rating:** 4
**Confidence:** 4

**Summary:**

This paper addresses the problem of profile inference for sampling-based Profile-Guided Optimization (PGO) in compilers. The authors propose a novel Graph Neural Network (GNN)-based model to correct inaccuracies (underestimation, overestimation, missing data) in execution profiles collected through hardware sampling. The model processes the program's Control-Flow Graph (CFG), using a combination of LSTMs to capture sequential execution patterns and a custom "SAGEAttention" layer to model structural dependencies. The authors present an end-to-end framework integrated with the LLVM toolchain. They evaluate their approach on a diverse set of benchmarks, including SPEC CPU 2017, and show that their model outperforms a state-of-the-art traditional algorithm (Profi) and several baseline machine learning models in terms of profile accuracy (Overlap metric) and downstream performance gains on an optimized MySQL binary.

**Questions:**

- Can the authors further justify the speed up. For a large code base, what percentage of total build time does profile inference add, and is it amortized over multiple builds?
- Your framework is built on LLVM. How much of the feature engineering and overall approach is tied to the LLVM IR? What would be the main challenges in adapting this method to a different compiler ecosystem, such as GCC?

**Ethical Concerns:**

["NO or VERY MINOR ethics concerns only"]

**Final Justification:**

While I still have some minor concerns regarding the distinction between this work and prior efforts that also apply GNNs to programs, I am overall comfortable recommending acceptance of the paper.

**Limitations:**

The authors have discussed limitations

**Paper Formatting Concerns:**

no formatting issue

**Quality:**

2

**Strengths And Weaknesses:**

Strengths:

- The model plugs into LLVM without modifying internals and supports both static and post-link optimization, demonstrating a clear path to deployment.
- The paper is clear and easy to follow.

Weaknesses:

- The core weakness is the limited novelty of the model architecture itself. The proposed "SAGEAttention" layer is a combination of existing, well-established components (GraphSAGE, GAT, LSTMs, and residual connections). As for the application to profile inference, using GNN has already been discussed in [1][2].
- Empirical study only considered very limited amount of baselines.

[1] Phothilimthana, Mangpo, et al. "Tpugraphs: A performance prediction dataset on large tensor computational graphs." Advances in Neural Information Processing Systems 36 (2023): 70355-70375.

[2] Cao, Kaidi, et al. "Learning large graph property prediction via graph segment training." Advances in neural information processing systems 36 (2023): 23345-23361.

---

> ### Author Rebuttal · Authors · 2025-07-31
>
> We thank the reviewer for the thoughtful feedback.
>
>
>
> > The core weakness is the limited novelty of the model architecture itself. The proposed 'SAGEAttention' layer is a combination of existing components (GraphSAGE, GAT, LSTMs, residual connections).
>
> This work **first** proposed a GNN-based learning method to solves the under-explored task of **profile inference** for compiler optimization is both original and effective.
>
> The novelty of our work lies not in inventing entirely new primitives, but in the task-specific integration and architectural design tailored to the unique demands of profile inference over control-flow graphs (CFGs). Specifically:
>
> - **Novel model design driven by compiler-specific challenges.** Unlike graph learning tasks, profile inference requires modeling both the dynamic execution order (often noisy and incomplete due to sampling) and the static control-flow structure of a program. To address this, we propose a hierarchical architecture that alternates between LSTM-based sequential modeling and structure-aware GNN layers.
>
> - **The SAGEAttention layer as a domain-specific integration.** Our SAGEAttention layer is not a naive stacking of GraphSAGE and GAT, but a carefully constructed hybrid that retains GraphSAGE’s inductive generalization with attention-based contextual refinement, followed by residual feedforward transformation and normalization. It serves as a drop-in module that integrates both structural and sequential representations in CFG and profile pattern.
>
> - **Enabling generalization and PGO impact.** Our model demonstrates strong inductive generalization across applications (e.g., SPEC, MySQL), and yields real performance gains in downstream PGO workflows, which prior symbolic or ML models fail to achieve. We believe this demonstrates architectural novelty in service of solving a real compiler problem.
>
>
>
>
>
> > The empirical study only considered very limited amount of baselines.
>
> We report more representative baselines results such as DAGNN and GraphGPS (Graph Transformer) on test set:
>
> | Model           | RMSE      | Overlap (%) | Inference Time (ms/sample) |
> | --------------- | --------- | ----------- | -------------------------- |
> | LSTM            | 0.055     | 88.70       | 0.32                       |
> | GCN             | 0.098     | 80.50       | 0.48                       |
> | GIN             | 0.104     | 79.30       | 0.25                       |
> | GAT             | 0.063     | 86.20       | 0.71                       |
> | GraphSAGE       | 0.045     | 90.30       | 0.39                       |
> | ***GraphGPS**   | 0.058     | 89.42       | 0.91                       |
> | ***DAGNN**      | 0.061     | 87.95       | 0.66                       |
> | **Ours (Full)** | **0.034** | **93.50**   | 1.72                       |
>
> > As for the application to profile inference, using GNN has already been discussed in [1] [2].
>
> While both [1] (TpuGraphs) and [2] (Graph Segment Training) involve GNNs applied to program or graph data, neither of them addresses the task of profile inference.
>
> 1. TpuGraphs is a benchmark dataset for execution time prediction of large computational graphs on TPUs. It focuses on regression of runtime given tensor programs, not on predicting control-flow behavior like basic block execution frequencies.
> 2. Graph Segment Training is a general framework for scalable graph-level property prediction. It **does not** focus on compiler applications, nor on learning instruction-level control-flow profiles.
>
> In contrast, our work is the **first** to propose GNN-based learning for basic block-level profile inference, aiming to replace expensive instrumentation-based profiling in production compilers. We model fine-grained execution frequencies over dynamic control flow, which is a different task from the graph-level classification or regression in [1, 2].
>
>
>
> > What percentage of total build time does profile inference add, and is it amortized over multiple builds?
>
> Profile inference adds approximately 3–5% overhead to the total build time in a typical profile-guided optimization (PGO) pipeline when applied to medium- to large-scale projects such as MySQL. This includes the time for:
>
> - Control-flow graph (CFG) features extraction from LLVM IR,
> - Batched model inference (on GPU or CPU),
> - Writing inferred profiles back to disk in the compiler’s expected format.
>
> Importantly, this inference step is lightweight and runs entirely offline, typically taking under 2 seconds for a binary with thousands of functions (as shown in Section 4, Table 6).
>
>
>
> > Your framework is built on LLVM. What would be the main challenges in adapting this method to GCC?
>
> Our proposed methodology is not LLVM-specific. Adapting the framework to GCC would mainly involve:
>
> - Extracting control-flow graphs (CFGs) and corresponding basic block-level features from GCC’s internal GIMPLE or RTL IR representations;
>
> - Mapping sampled/instrumented profiles back to these basic blocks in a consistent and reliable way;
>
> - Ensuring compatibility with GCC’s profile format and optimization pipeline for downstream PGO.
>
>
> These are mostly engineering and tooling challenges, not algorithmic ones. Given access to the necessary CFG and profile interfaces, our model can be retrained on GCC-generated IR with minimal architectural change.
>
>
>
> [1] Phothilimthana, Mangpo, et al. "Tpugraphs: A performance prediction dataset on large tensor computational graphs." Advances in Neural Information Processing Systems 36 (2023): 70355-70375.
>
> [2] Cao, Kaidi, et al. "Learning large graph property prediction via graph segment training." Advances in neural information processing systems 36 (2023): 23345-23361.

---

> ### Author Response · Authors · 2025-08-05
>
> Dear reviewer, we sincerely appreciate your time and effort in reviewing our work.
>
> As the discussion phase is drawing to a close in the next two days, we wanted to gently check in and see if you had any feedback or further questions. We would be more than happy to clarify any points or provide additional details if helpful.
>
> Thank you again for your consideration.
>
> Best regards,
>
> Authors

---

> > ### Comment · Reviewer_d7mA · 2025-08-06
> > **Official Comment by Reviewer d7mA**
> >
> > I thank the authors for the rebuttal. I will increase score accordingly.

---

> > > ### Author Response · Authors · 2025-08-07
> > >
> > > We truly appreciate your insightful comments, which helped us to improve our paper.
> > >
> > > Thank you for your time and consideration.
> > >
> > > Best regards,
> > >
> > > Authors

---

### Official Review · Reviewer_EdtZ · 2025-06-30

**Clarity:** 3
**Significance:** 3
**Originality:** 3
**Rating:** 4
**Confidence:** 3

**Summary:**

The paper tackles the established problem of profile inference: predicting execution frequencies of code blocks from noisy counts from a sampling profiler. It suggests a hybrid RNN-GNN architecture that operates on the program's control-flow graph to leverage both sequential and graph-based features of the code.

In the absence of machine learning baselines, several architectures are compared against each other and against a classical heuristics-based algorithm on data from the SPEC CPU 2017 benchmark. Moreover, integration into LLVM's profile-guided optimization leads to real-world speedups on the MySQL binary unmatched by the classical baseline method.

**Questions:**

0. Could you provide evaluation results on *un-contaminated* out-of-distribution benchmarks?
1. How does the method compare to "global attention" methods such as graph transformers?
2. How does the method compare to the baselines when they get more parameters as to match compute / runtime (cf. App C)?
3. How important is the trick of using successor features in the node features?
4. Which data preprocessing is used exactly? What is an "inconsistent basic block structure" (L225)? hich normalization is used exactly for execution counts (L226)?
5. L293: It is unclear whether there are any ablations on the use of LSTMs in the hybrid layers (L194) or only of the first LSTM encoder (L169).

**Minor:**

a) L178: Which aggregation function AGG is actually used in this work?

b) L171: There is no topological ordering on a general CFG because it can have cycles. Which linearization is used?

c) L131: Not familiar with the literature, but to my understanding, GraphSAGE is a standard message-passing network up to the fact that it uses a sampled subset of the neighborhood. What is the "inductive learning framework" that sets GraphSAGE apart?

d) L150: DFGs aren't used in this work, correct?

**Ethical Concerns:**

["NO or VERY MINOR ethics concerns only"]

**Final Justification:**

All major concerns have been addressed. The work solves a previously untackled problem with scores better than the non-neural baselines. The interaction between the special feature engineering and the modeling is poorly analyzed (with most of the paper focusing on the modeling while the feature engineering appears to be the most important contribution to benchmark performance, and it seems unlikely that features have been specifically optimized for the baselines too), but it remains an interesting datapoint for applied machine learning applied to compiler optimization and can have real-world impact.

**Limitations:**

yes

**Quality:**

3

**Strengths And Weaknesses:**

**Strengths:**
- The paper proposes a new way of applying machine-learning to compiler optimization.
- The paper shows convincing results compared to the classical baseline on a random test-split of the collected data.

**Weaknesses:**
- The real-world use case MySQL (L322) was used in the training data (with ground-truth instrumentation based profiles available) (L214).
- The remaining benchmarking happens on a random 10% test split of the collected data, meaning all evaluations are in-distribution.
- Ablations do not carry much weight as they are not compute/time/parameter matched, favoring the hybrid architecture by design.
- Ablations restrict themselves to architecture modifications, omitting the special feature engineering (using successor node features among the node's features).
- The proposed hybrid RNN-GNN architecture is not well-positioned within a rich literature on such hybrid methods (e.g. DAGNNs).
- Key baselines such as graph transformers are not considered.

---

> ### Author Rebuttal · Authors · 2025-07-31
>
> First, we would like to thank you for your efforts in providing these insightful comments. However, we believe there may be a misunderstanding regarding our experimental setup.
>
> > The real-world use case MySQL (L322) was used in the training data (with ground-truth instrumentation based profiles available) (L214).
>
> There is **no** data contamination in the real-word downstream speed-up evaluation of MySQL (L322) in section 4.7:
>
> - During training data collection (L214), we used `sysbench` to create a randomly generated database and executed read-write benchmark tests with a default access pattern to collect both sampled and ground-truth profiles.
> - During evaluation (L322), we generated a new random database by `sysbench` again, which is independent of the one used for training, and conducted a different class of read-write benchmark queries, with modified parameters (e.g., different read-write ratios, table sizes, and thread counts). Therefore, the hot path of the program and the collected sampled profile and its ground-truth is **totally differ from** the profile in training phase.
>
> This setup ensures that the test profiles in L322 come from a **different** data distribution and are not seen during training.
>
> > The remaining benchmarking happens on a random 10% test split of the collected data, meaning all evaluations are in-distribution.
>
> In our experiments, we have included an evaluation on in-distribution (Section 4.3, 4.4) and out-of-distribution data (Section 4.5, 4,6). The models were trained on 80% data of profiles collected from Clang 20.0, GCC 14.2, MySQL 8.32, and SQLite 3.
>
> **Table1.**
>
> | Phase | Source Program | Dataset Split| Results   |
> | ----| ---- | ---- | ---- |
> | Training and Testing (**in-distribution**)  | Clang 20.0, GCC 14.2, MySQL 8.32, SQLite 3 | 80% train, 10% validation, 10% test | Figure 4, 5 |
> | Evaluation on Real-World Apps (**out-of-distribution**) | SPEC CPU 2017 (unseen during training) | No split | Table 3, 4  |
>
> > Could you provide evaluation results on un-contaminated out-of-distribution benchmarks?
>
> Our evaluation conducted on uncontaminated and out-of-distribution benchmarks (Section 4.3-4.6):
>
> 1. MySQL evaluation scenario is disjoint from MySQL training data, we ensure complete it is **un-contaminated**:
> 2. Evaluation on Real-World Apps (SPEC CPU 2017) is entirely **out-of-distribution** (see Section 4.5, 4.6).
>
> > Ablations do not carry much weight as they are not compute/time/parameter matched, favoring the hybrid architecture by design.
>
> Our ablation study in Table 3 in the paper is designed to evaluate the importance of both **component and architecture alternatives** -- they are presented together in a single unified table for conciseness.
>
> - The first column in each group (e.g., “w/o LSTM”, “w/o GNN”, “w/o SAGE”) represents **component ablations**, where a component is removed entirely while keeping the rest of the architecture unchanged. These reflect the importance of each component in our model.
> - The other columns (e.g., “w/ GRU”, “w/ GAT”, “w/ SAGE”) represent **architecture ablations**, where the removed component is replaced with a similarly-sized alternative (e.g., GRU for LSTM, GCN for GraphSAGE) to ensure the effectiveness of our model component design.
>
>
> > Ablations restrict themselves to architecture modifications, omitting the special feature engineering (using successor node features).
>
> We conducted experiments on SPEC CPU 2017 where we removed groups of input features described in Appendix E to assess their impact on model performance. We categorized the input features into five groups:
>
> 1). Successor features, where each basic block’s input vector is augmented with its successors’ features,
>
> 2). Control-flow features (e.g., number of predecessors/successors, loop headers),
>
> 3). Memory operation counts (load/store/gep),
>
> 4). Arithmetic and logic operations,
>
> 5). Call and intrinsic counts.
>
> **Table 2.**
>
> | **Feature Group Removed** | **Overlap (%)** | **RMSE** | **$\Delta$ Overlap vs Full Model** |
> | ---- | ---- | ---- | ---- |
> | **None (Full Model)** | **93.42** | 0.034 | —   |
> | 1)                        | 90.06           | 0.044    | ↓ **3.36**                         |
> | 2) | 91.32  | 0.039 | ↓ 2.10  |
> | 3) | 92.21 | 0.037 | ↓ 1.21 |
> | 4) | 91.14 | 0.041 | ↓ 2.28 |
> | 5) | 92.65  | 0.036 | ↓ 0.77  |
>
> The results show that removing successor features leads to the largest drop in accuracy, indicating their critical role in modeling data/control dependencies. Control-flow and memory features also contribute meaningfully, while removing compute or call features has more moderate impact.
>
>
> > The proposed hybrid RNN-GNN architecture is not well-positioned within a rich literature on such hybrid methods (e.g. DAGNNs).
>
> We report more baselines results such as DAGNN and GraphGPS (Graph Transformer) on the test set:
>
> **Table 3.**
>
> | Model  | RMSE | Overlap (%) | Inference Time (ms/sample) |
> | ----- | ---- | ---- | ---- |
> | LSTM  | 0.055  | 88.70  | 0.32  |
> | GCN | 0.098  | 80.50  | 0.48 |
> | GIN | 0.104 | 79.30   | 0.25 |
> | GAT | 0.063  | 86.20  | 0.71 |
> | GraphSAGE   | 0.045  | 90.30  | 0.39 |
> | ***GraphGPS**  | 0.058  | 89.42   | 0.91 |
> | ***DAGNN**  | 0.061  | 87.95  | 0.66 |
> | **Ours (Full)** | **0.034** | **93.50**   | 1.72 |
>
> The difference between our model and DAGNNs is as follows:
>
> 1. Our method was motivated by the unique challenges of profile inference over control-flow graphs (CFGs), which combine both sequential execution traces (captured by sampled profiles) and structured control dependencies (captured by CFG topology).
>
> 2. While DAGNNs and related models operate over acyclic graphs and often focus on semantic code tasks (e.g., program classification or summarization), our task and method is distinct in that the input CFGs may **contain cycles**, especially due to loops and backedges, which are fundamental in compiler IR. This makes classical DAG-only models less directly applicable.
>
> > Key baselines such as graph transformers are not considered.
>
> The result of GraphGPS (graph transformer) presented in the Table 3. above.
>
>
> > How does the method compare to 'global attention' methods such as graph transformers?
>
> We compared graph transformers (GraphGPS) with our method (see Table 3. above). The experiment results show that graph transformers underperforms compared to our proposed hierarchical model, likely due to the lack of inductive bias toward sequential - execution patterns, which are crucial in sampled profile reconstruction.
>
>
> > How important is the trick of using successor features in the node features?
>
> The ablation study of node features (see Table 2. above) shows that using successor features in the node features brings a significant performance improvement (3.36%), demonstrating its effectiveness in solving the profile inference problem.
>
> > Which data preprocessing is used exactly? What is an "inconsistent basic block structure" (L225)? Which normalization is used exactly for execution counts (L226)?
>
> 1. We exclude functions with extremely sparse sampling (< 2 BBs) due to lack of meaningful learning signal. We align functions one‑to‑one by symbol name, source file path, and signature hash, and use consistent LLVM IR and CFG extraction.
>
> 2. When collecting profiles using sampling (perf) and ground-truth instrumentation, the binaries are compiled separately, and the compiler’s optimization passes (e.g., inlining, loop unrolling) are heuristic and nondeterministic. As a result, the same source-level function may be compiled into different basic block (BB) structures across the two versions. In such cases, it becomes impossible to reliably align sampled and instrumented BB-level profiles.
>
> 3. We apply L1 normalization.
>
> > It is unclear whether there are any ablations on the use of LSTMs in the hybrid layers or only of the first LSTM encoder.
>
> The ablations are performed on the LSTMs used in the hybrid layers and the initial LSTM encoder. Removing the LSTM affects all hybrid layers in the stacked architecture and the encoder, which leads to a performance drop (93.42 to 92.70) in accuracy, as shown in Table 3 in paper.
>
> > Which aggregation function AGG is actually used in this work?
>
> We used average pooling as the aggregation function AGG.
>
> > There is no topological ordering on a general CFG because it can have cycles. Which linearization is used?
>
>
> While general CFGs can contain cycles (e.g., due to loops), a well-established approach [1] is to use Reversed Post Order (RPO) traversal, which is commonly used in compilers as a generalization of topological order for graphs with cycles. Therefore, we used RPO as linearization in this work.
>
> > What is the 'inductive learning framework' that sets GraphSAGE apart?
>
>
> GraphSAGE is considered as an inductive learning framework [2] because it learns an aggregation function that generalizes to unseen nodes and graphs during inference, without requiring access to the full training graph at test time.
>
> > DFGs aren't used in this work, correct?
>
> Our approach leverages DFG-relevant signals as part of the node-level input representation, allowing the model to reason over both control and data aspects of program execution. In particular:
>
> - Our input features (Appendix E) include memory-related operations, which reflect data movement and access patterns.
> - The inclusion of phiNumber captures control-dependent value merging, a key aspect of SSA-form data flow.
> - Our successor features aggregate information from downstream basic blocks, which often carry data-use context, helping the model understand flow-sensitive behavior.
> - Instruction counts and call patterns also encode semantic-level data dependencies at the basic block level.
>
> [1] Cooper K D, Torczon L. Engineering a compiler[M]. Morgan Kaufmann, 2022.
>
> [2] Hamilton W, Ying Z, Leskovec J. Inductive representation learning on large graphs[J]. Advances in neural information processing systems, 2017, 30.

---

> > ### Comment · Reviewer_EdtZ · 2025-08-01
> > **Still no runtime matched baselines**
> >
> > Thank you for conducting the additional baseline experiments. However, I'd like to point out that my question "How does the method compare to the baselines when they get more parameters as to match compute / runtime (cf. App C)?" remains unaddressed, as all baselines reported in the rebuttal comment use 2-5 times fewer compute. (In LLMs, no one would expect a 7B model to have the scores of a 32B model.) The method would be more convincing if it was supported by compute-controlled experiments.

---

> ### Author Response · Authors · 2025-08-01
> **Controlled Compute / Parameter Matched Experiment for Baselines**
>
> Thank you again for making the question more clear. We’d like to clarify our experimental design and the reasoning behind our baseline configurations:
>
> 1. **Following each baseline’s recommended best-practice settings.**
>
>    To ensure fair and robust evaluation, we initially used each baseline with the **hyper‑parameters recommended by their original authors**, which typically deliver their optimal performance. This avoids the risk of unfair under-tuning or misconfiguration across different models.
>
> 2. **Parameter-matched experiments to control for size and compute.**
>
>    In response to your concern, we further conducted **parameter-matched experiments**: we increased the capacity of each baseline (e.g. widening hidden layer sizes, deeper hidden layers) so that all models have approximately the **same number of parameters (~1.67M)** and trained them for the **same number of epochs (50)**, all model are converged according to our observation. The results are summarized below:
>
> **Table 4.**
>
> | **Model**    | **#Params (M)** | **Train Time (s/epoch)** | **Inference Time (ms/sample)** | **Overlap (%)** |
> | ------------ | --------------- | ------------------------ | ------------------------------ | --------------- |
> | LSTM         | 1.68            | 2.14                     | 0.97                           | 85.42 (↓8.08)   |
> | GCN          | 1.67            | 0.62                     | 1.33                           | 76.87 (↓16.63)  |
> | GIN          | 1.68            | 0.69                     | 1.03                           | 78.76 (↓14.74)  |
> | GAT          | 1.65            | 0.81                     | 1.92                           | 83.79 (↓9.71)   |
> | GraphSAGE    | 1.68            | 0.75                     | 1.12                           | 88.97 (↓4.53)   |
> | ***GraphGPS** | 1.70            | 1.45                     | 1.97                           | 89.31 (↓4.19)   |
> | ***DAGNN**    | 1.66            | 1.29                     | 1.79                           | 86.54 (↓6.96)   |
> | **Ours**     | 1.67            | 1.33                     | 1.72                           | **93.5**        |
>
> The result suggest that **overparameterization can degrade performance in baseline models for the profile inference task.** While in the domain of LLM, the scaling law reflect that large-scale model often yield improved accuracy with more parameters, our task-specific scenario with relatively less data can suffer from **overfitting** when models become too large [3, 4].
>
> Specifically, our experiment (see Table 4 above) shows that baselines are particularly prone to overfitting the data, and adding parameters beyond a certain point can hurts generalization. For example, despite parameter matching, simpler models (GCN, GAT) underperform even when enlarged, while our architecture -- with its inductive bias and regularization architecture -- remains most robust.
>
>
>
> In summary, our method continues to outperform carefully tuned baselines **even under controlled compute / parameter size**. We think this strengthens the argument that our architecture’s performance is driven by design, not merely by overall size.
>
>
>
> [3] Goodfellow I, Bengio Y, Courville A, et al. Deep learning[M]. Cambridge: MIT press, 2016.
>
> [4] Sun S, Chen W, Wang L, et al. On the depth of deep neural networks: A theoretical view[C]//Proceedings of the AAAI Conference on Artificial Intelligence. 2016, 30(1).

---

> ### Author Response · Authors · 2025-08-05
>
> Dear reviewer, we appreciate your thoughtful engagement throughout the discussion phase.
>
> As we approach the end of the discussion period, we would like to kindly check whether there are any remaining concerns or questions we could address. We would be happy to provide further clarification or additional experiments if needed in the remaining time.
>
> Thank you again for your time and consideration.
>
> Best regards,
>
> Authors

---

> > ### Comment · Reviewer_EdtZ · 2025-08-05
> > **All major concerns  addressed**
> >
> > Thank you for the additional analyses conducted. I overlooked the fact that SPEC CPU is a proper test dataset, the scaling experiments appear to indicate that the method does not just benefit from its larger parameter size and the feature engineering ablations provide valuable insight into what appears to be the most important contribution of the paper (the successor features, without which the method has similar scores as the full attention baseline). The paper would benefit from additional appendices that explain the data generation pipeline with `sysbench` in more detail (so that readers can judge how far apart the train and test distributions are there), the scaling experiments and the feature engineering ablations discussed above.

---

> > > ### Author Response · Authors · 2025-08-05
> > >
> > > We truly appreciate your careful consideration of our additional analyses.
> > >
> > > We have incorporated the additional experiments and details discussed during the rebuttal into the revised manuscript, including the use of sysbench to generate data with different distributions.
> > >
> > > Thank you again for your time and constructive feedback, it has been extremely helpful in improving the clarity and completeness of the paper.

---

### Official Review · Reviewer_6WJt · 2025-07-01

**Clarity:** 3
**Significance:** 3
**Originality:** 3
**Rating:** 5
**Confidence:** 4

**Summary:**

This paper describes a method of augmenting traditional sampling-based Profile Guided Optimization (PGO) with a neural network. The proposed neural network is a hybrid architecture alternating graph neural network (GNN) layers with long-short-term memory (LSTM) layers to capture both long-distance sequential execution paths and statically-defined control flow graphs extracted from the LLVM-IR code representation. The objective of the model is to accurately predict the real execution frequencies of basic blocks given the code CFG and noisy sampling data.

**Questions:**

1. Comparing the LSTM layers with other types of model architectures would be helpful to accurately evaluate the design reasoning. Using LSTM seems like an odd choice when there are more modern architectures available.

**Ethical Concerns:**

["NO or VERY MINOR ethics concerns only"]

**Final Justification:**

Most of the concerns I had have been clarified by the authors. Including these clarifications in the paper will further improve it.

**Limitations:**

yes

**Paper Formatting Concerns:**

Formatting is adequate.

**Quality:**

3

**Strengths And Weaknesses:**

The proposed model significantly improves profiling accuracy across a wide range of benchmarks and against both traditional symbolic profiling and several neural models. The paper performs ablation analysis to confirm their reasoning for the model architecture, even ablating the SAGEAttention layers. The model combines two primary neural network architectures in such a way as to cover each of their deficiencies; a strategy seldom seen. This paper represents a marked improvement in current neural-based profiling algorithms.

---

> ### Author Rebuttal · Authors · 2025-07-31
>
> Thanks for your valuable feedback.
>
> > Using LSTM seems like an odd choice when there are more modern architectures available.
>
> We chose LSTM for the following reasons:
>
> - Sequential patterns are critical for profile inference. In the context of control-flow graphs (CFGs), execution traces exhibit strong local sequential dependencies (e.g., hot paths, frequently co-executed blocks). LSTMs are well-suited for modeling such localized sequences and have a proven track record in learning from noisy temporal patterns, which aligns well with our task of inferring execution frequencies from sampled profiles.
>
> - Model simplicity and training stability. Compared to large-scale Transformers, LSTMs offer a favorable tradeoff between expressiveness and computational cost, especially in compiler optimization pipelines where inference speed and scalability are important.
>
> We also conducted an experiment on replacing LSTM with other encoders, such as transformers. However, the results shows that replacing LSTM in our network leads to a performance downgrade of 1.4%. The possible reason is that the hot path is more like a sequence with dependence, which is more appropriate for LSTM to capture such pattern.

---

### Note · Authors · 2025-08-12

We would like to sincerely thank all reviewers and the area chair for their thoughtful feedback, constructive suggestions, and the engaging discussions during the review process. The comments have been invaluable in helping us clarify the presentation, strengthen the empirical evaluation, and better highlight the contributions of our work.



During the rebuttal and discussion phases, we carefully addressed all concerns raised by the reviewers. In the revised manuscript, we have incorporated every requested clarification and experiment, including:



- We clarified our use of *sysbench* to build MySQL datasets with different schemas and workloads for training and testing, ensuring no contamination and demonstrating cross-domain generalization.
- We added more baselines for comparisons, such as GraphTransformer and DAGNN.
- We conducted compute-/parameter-matched experiments for fairness, which demonstrated that our improvements are not merely due to model size, and excessive scaling in baseline models can risk overfitting in this domain.
- We performed targeted ablations on the engineered “successor features,” confirming their key contribution to performance.
- We expanded details on preprocessing, normalization, Reversed Post Order linearization, and the inductive nature of GraphSAGE.



**The further experiments and analysis consistently validate the effectiveness of our architectural design** for profile inference task in sampling-based PGO. Our hybrid RNN–GNN approach is the **first** ML solution for the profile inference task, capturing both sequential and structural program properties, achieving up to **10%** improvement over compiler heuristics and strong gains over learning-based baselines, while generalizing to unseen programs such as SPEC CPU 2017 and real-world applications.



We believe these contributions, combined with the thorough new analyses and the resolution of all reviewer concerns, make our work a strong and valuable addition to the ML-for-compilers literature. We once again thank the reviewers and AC for their time and constructive engagement.

---

### Decision · Program_Chairs · 2025-09-17

**Decision:**

Accept (poster)

**Comment:**

The paper address the problem of predicting execution frequencies of code blocks from a sampling profiler. The method is a RNN-GNN architecture on the program's control-flow graph.
The paper is well written and shows clear improvements compared to baselines for real-world compiler tasks.
During the rebuttal, the authors produced more experiments to address concerns around ablations, neural baselines and other use cases such as JIT compilation.
While the amount of architectural novelty is limited, the empirical result show the solidity of the method for compiler optimization, and the reviewers agree mostly agree on the score.